# Improving Image Captioning via Predicting Structured Concepts

**Ting Wang**[1], **Weidong Chen**[1*], **Yuanhe Tian**[2], **Yan Song**[1], **Zhendong Mao**[1]

[1] University of Science and Technology of China
[2] University of Washington

wt1023@mail.ustc.edu.cn, {chenweidong, zdmao}@ustc.edu.cn,
yhtian@uw.edu, clksong@gmail.com

## Abstract

Having the difficulty of solving the semantic gap between images and texts for the image captioning task, conventional studies in this area paid some attention to treating semantic concepts as a bridge between the two modalities and improved captioning performance accordingly. Although promising results on concept prediction were obtained, the aforementioned studies normally ignore the relationship among concepts, which relies on not only objects in the image, but also word dependencies in the text, so that offers a considerable potential for improving the process of generating good descriptions. In this paper, we propose a structured concept predictor (SCP) to predict concepts and their structures, then we integrate them into captioning, so as to enhance the contribution of visual signals in this task via concepts and further use their relations to distinguish cross-modal semantics for better description generation. Particularly, we design weighted graph convolutional networks (W-GCN) to depict concept relations driven by word dependencies, and then learns differentiated contributions from these concepts for following decoding process. Therefore, our approach captures potential relations among concepts and discriminatively learns different concepts, so that effectively facilitates image captioning with inherited information across modalities. Extensive experiments and their results demonstrate the effectiveness of our approach as well as each proposed module in this work. Source code is available at: https://github.com/wangting0/SCP-WGCN.

## 1 Introduction

The image captioning task aims at generating a human-like description for a given image, normally requiring recognition and understanding of the content in the image, including objects, attributes, and their relationships, etc. The task is regarded as

---

*Corresponding author: Weidong Chen.

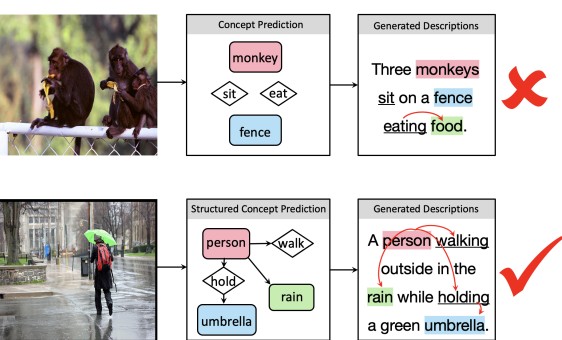

Figure 1: Illustrations of our motivation. Compared with integrating semantic concepts into the image captioning framework, we find that the structured concepts helps reduce over-reliance on linguistic priors in language generation.

an interdisciplinary research of computer vision and natural language processing and has become a popular topic in recent years.

Current methods (Vinyals et al., 2015; Anderson et al., 2018; Huang et al., 2019; Cornia et al., 2020; Pan et al., 2020; Luo et al., 2021; Fang et al., 2022; Yang et al., 2022; Wu et al., 2022) usually follow an encoder-decoder framework, using a pre-trained object detector/classifier as an encoder to mine visual information in an image, and then feeding it into an RNN- (Zaremba et al., 2014) or Transformer- (Vaswani et al., 2017) based decoder for description prediction along with partially generated words. However, in most cases, the extracted visual information is insufficient, even with the use of powerful visual feature extractors. This shortcoming makes the decoder rely too much on partially generated words to predict the remaining words to ensure the fluency of the generated description, that is, the model relies too much on linguistic priors during decoding, and sometimes the resulted words does not related to the image at all. In short, the major challenge that the image captioning facing now is that description generation relies too much on linguistic priors and has little to do with images.

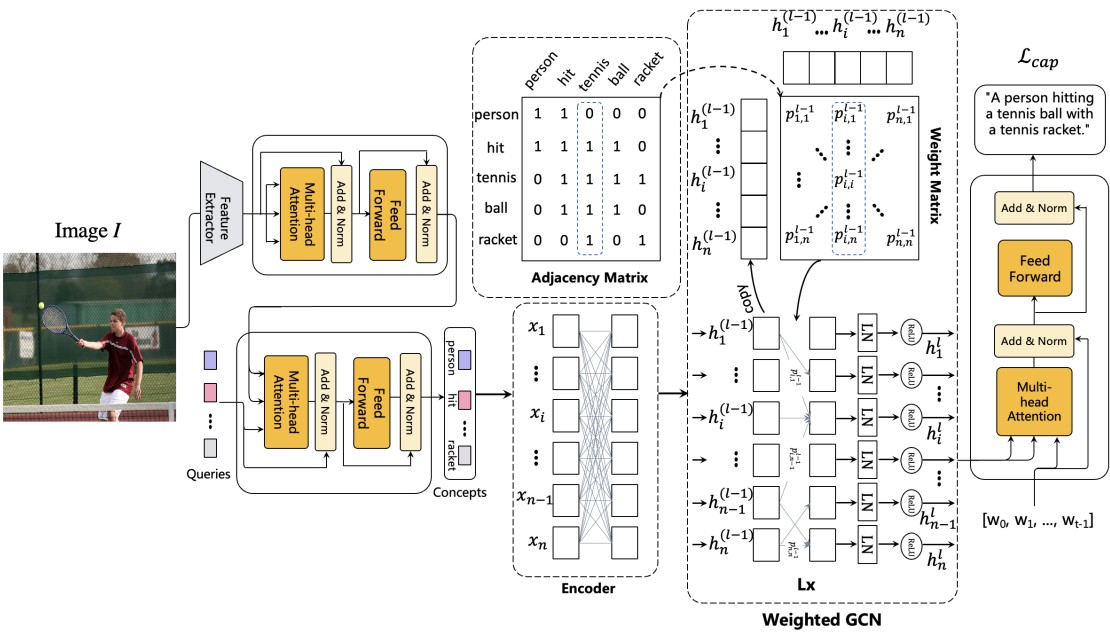

Figure 2: The overall framework of our approach, including concept prediction, weighted graph convolutional networks (W-GCN) and language decoder. The last weight matrix refers to the relations between concepts. Our model can be trained in an end-to-end manner.

To this end, some studies(Wu et al., 2022) add additional visual information to obtain strong visual features and increase the contribution of vision to generate captions. The other studies (Fang et al., 2022; Gan et al., 2017; Bin et al., 2017; Gao et al., 2020) have noticed the importance of semantic concepts based on visual content, which can provide rich and accurate semantic understanding, accordingly helping semantic alignment and generating reliable text descriptions. Thus the conventional pipeline of their studies is to firstly predict the semantic concepts, and then send the predicted semantic concepts along with the visual features into the decoder to predict description. Although promising results are obtained by these studies, they still ignore the relations among predicted concepts, which would not alleviate the overfitting on linguistic priors. For example, in one possible scenario that a model predicts the semantic words "baby" and "drink" from an image, the model is much more likely to predict the sentence as "baby → drink → milk" if it understands the relations between "baby" and "drink" is "baby → drink". On the contrary, "drink" is more likely to be followed by "water" based on the linguistic prior if the model ignores the relations. Another example is shown in Figure 1, the predicted concept itself makes it difficult for language generation to escape linguis-

tic priors. Meanwhile, it is seen that concepts have relations, which are not only shown among objects in the image, but also in word dependencies in the text. Structured semantic concepts improve the performance of the model from the perspective of language generation.

In this paper, we propose a structured concept predictor (SCP) to improve image captioning, which not only integrates concept prediction into the end-to-end image captioning, but also predicts the structures of obtained concepts based on the word dependencies. Specifically, we propose weighted graph convolutional networks (W-GCN), with its input graph built based on mutual information priors of all descriptions in an unsupervised manner. The mutual information priors are the probability of co-occurrence of two words within a certain distance in the same description, making the language generation no longer limited to local contextual information and measuring the relationship between two concept words well. Meanwhile, applying attention over the graph makes the concept features discriminatively learned, thereby reducing linguistic priors in captioning. Thus, we structuralize the semantic concepts and integrate them into the end-to-end image captioning, which keep the description generation associated with image at all generation steps and reduce over-reliance on lin-

guistic priors. Experiments on the widely used MS COCO benchmark demonstrate the superiority of our approach over strong baselines, with qualitative analyses confirming its ability in better capturing structural relationship among semantic concepts, so that offers good interpretability for our method.

## 2 The Approach

In this section, we describe our proposed SCP, which extracts concepts from a given image and utilizes graph convolutional networks to build topological structures between concepts, so that the structured concepts can help generate descriptive text. Figure 2 gives an overview of SCP, which consists of three main components.

### 2.1 Visual Feature Processing

**Extractor.** To generate descriptions, the first step is to extract the visual features from images. Following (Li et al., 2022b), in order to narrow the semantic gap between images and text, and help semantic alignment, we extract visual features from images $I$ by using the encoder of CLIP (Mokady et al., 2021) with the ResNet-101 (He et al., 2016) backbone, which has the ability to understand complex scenarios, and having excellent domain generalization ability after pre-training with a large dataset. The process can be formulated as:

$$X = f_v(I) \tag{1}$$

where $f_v$ is the visual extractor, and $X$ is the image features.

**Encoder.** Since the image feature is in the form of 2D, we first flatten $X$ into a sequence $\{x_1, x_2, ..., x_S\}$, $x_s \in \mathbb{R}^d$, where $x_s$ are patch features and $d$ is the size of the feature vector. Then we employ $N_v$ Transformer encoder blocks to further encode image features as a sequence. Outputs are the hidden states encoded from the input features $X$ extracted from the visual extractor. The whole process can be formulized as follows:

$$H_v^{(0)} = X \tag{2}$$

$$\overline{H}_v^{(l)} = \text{MHA}(H_v^{(l-1)}, H_v^{(l-1)}, H_v^{(l-1)}) \tag{3}$$

$$H_v^{(l)} = \text{LN}(\overline{H}_v^{(l)} + H_c^{(l-1)}) \tag{4}$$

where LN is the layer normalization, MHA stands for multi-head attention, $H_v^{(l)}$ indicates the output of the $l$-th middle hidden layer and the superscript indicates the number of layer. In particular, $H_v^{N_v}$ is the output of the Transformer encoder. For simplicity, let $\widetilde{V}$ denote $H_v^{N_v}$ in the rest of paper.

### 2.2 Concept Prediction

Most existing image captioning works leverage a pre-trained object detector to capture the semantics in an image, which are then directly fed into language decoder to generate the descriptive sentence. However, the semantic perception capability of the pre-trained detector is severely limited by pre-defined class labels. Based on the grid visual features, it predicts concepts from ground-truth captions through a multi-label classification task. These concepts contain rich, comprehensive, accurate and refined semantic information, which can be decoded directly by multi-modal decoders, encouraging the generation of relevant word at each decoding step, greatly benefiting the image captioning task.

Specifically, we predict the semantic concepts under the guidance of visual features through a set of concept queries $Q$, by leveraging $N_c$ Transformer encoder blocks. The set of learnable queries learns the essential concepts within the images. Through the image interaction, each of our learnable queries focuses on a specific area of the image and learns the information (concepts) contained in the image. These concepts include objects, relative positions between objects, actions, etc. Each Transformer block reinforces concept queries by interacting visual features $\widetilde{V}$ with object queries $Q$ through a cross-attention mechanism. The whole process can be formulized as follows:

$$H_c^{(0)} = Q \tag{5}$$

$$H_c^{(l)} = \text{LN}(\text{MHA}(H_c^{(l-1)}, \widetilde{V}, \widetilde{V}) + H_c^{(l-1)}) \tag{6}$$

where $H_c^{(l)}$ indicates the output of the $l$-th middle hidden layer and the superscript indicates the number of layer. Thus, the output of the Transformer encoder $H_c^{N_c}$ is the output of the last Transformer layer.

We then feed the output of the last block into a multi-linear perception network to get concept features $C$:

$$C = \text{MLP}(H_c^{N_c}) \tag{7}$$

where MLP is the multi-linear perception network with the sigmoid activation.

During training, following (Fang et al., 2022), we describe it as a multi-label classification problem. Due to the imbalance of the distribution of concepts, we use asymmetric loss (Ben-Baruch et al., 2020), which can handle the sample imbalance problem of multi-label classification tasks

well. Asymmetric loss is calculated for concept prediction:

$$\mathcal{L}_c = \textbf{asym}(C, Y_c) \qquad (8)$$

where $Y_c$ denote the visual concept of the ground-truth sentence that corresponds to the **concept vocabulary** (The details of building concept vocabulary are introduced in §3.2).

## 2.3 Weighted Graph Convolutional Network

After obtaining the enriched semantics derived from concept predictor, the most typical way to predict descriptions is to directly feed the semantic features, which is obtained by concept predictor, into the RNN/Transformer-based language decoder. However, in this way, the language decoder overly relies on the language priors to generate captions, since those concepts are treated independently, and their features are learned independently. A straghtforward example is illustrated in the introduction section.

To this end, we propose to construct graph for these concepts, explore the relationship between them by Weighted Graph Convolutional Networks (W-GCN), and obtain structured concepts. Structure concepts learns to estimate the linguistic relative position of semantic word pairs, thereby allocating all the semantic words in potential linguistic order as humans. In doing so, the output sequence of structured semantic concepts serve as additional visually-grounded language priors, which encourage the visual contribution in generation.

Concretely, the nodes of the graph represent concepts $G = \{g_1, g_2, ..., g_k\}$, and the edges represent the relationship between nodes $g_i$ and $g_j$ for $\forall i, j \in \{1, 2, ..., k\}$, which can be represented by an adjacency matrix $A$. In $A$, $a_{ij} = 1$, if there is an edge between $g_i$ and $g_j$ or $i = j$, otherwise $a_{ij} = 0$.

### 2.3.1 Graph Construction

As the obtained semantic concepts cannot form a complete sentence, our method cannot leverage existing dependency parsers to estimate their relations. Without such a parser, we need an alternative way to find satisfied word pairs to build initial graphs in our W-GCN, which equivalent to build the initial adjacency matrix $A$. Inspired by the studies (Tian et al., 2020) which leverage chunks (n-grams) as additional features to carry contextual information, we propose to construct the graph based on the word dependencies extracted from a pre-constructed n-gram lexicon $D$.

Specifically, we count the frequency of the occurrences of each word and the frequency of simultaneous occurrence of any two words within $N_L$ word distance (considering the order) in all sentences of the training set. We regard two words within $N_L$ distance as they have word dependency. Then we calculate the Pointwise Mutual Information (PMI) score of any two words $w_1$, $w_2$ by the following formula and set a threshold to determine if they are strongly correlated.

$$PMI(w_1, w_2) = log\frac{p(w_1 w_2)}{p(w_1)p(w_2)} \qquad (9)$$

where $p(w_1), p(w_2)$ is the probability of $w_1, w_2$ in the training set, $p(w_1, w_2)$ is the probability that both $w_1$ and $w_2$ are within $N_L$ word distance.

We store all strongly correlated word pairs in a word lexicon D and refer to it to build the graph. If the concept represented by the two nodes in the graph can be found in the lexicon D, then the corresponding element of its adjacency matrix is initialized as correlated, which is set to 1. Otherwise, setting the value to 0.

### 2.3.2 The Weighted GCN

Based on the adjacency matrix, the W-GCN module of the $L$ layers can learn from all the input concepts. Considering that the contribution of different $g_j$ to $g_i$ may be different, we further apply the attention mechanism to the adjacency matrix, replacing $a_{ij}$ with the weights $\alpha_{ij}$. For each $g_i$ and all its related $g_j$, we calculate weight $\alpha_{ij}$ for the concept pair. In particular, at the $l$-th layer, for each $g_i$, all the $g_j$ associated with it can be calculated:

$$\alpha_{ij}^{(l)} = \frac{a_{ij} \cdot exp(h_i^{(l-1)} \cdot W_{pos}^{(l)} \cdot h_j^{(l-1)})}{\sum_{j=1}^{n} a_{ij} \cdot exp(h_i^{(l-1)} \cdot W_{pos}^{(l)} \cdot h_j^{(l-1)})} \qquad (10)$$

$$h_i^{(l)} = \sigma(\text{LN}(\sum_{j=1}^{n} \alpha_{ij}(W^{(l)} \cdot h_j^{(l-1)} + b^{(l)}))) \qquad (11)$$

where $W_{pos}^{(l)}$ is a trainable parameter, it can model the position relationship between $g_i$ and $g_j$ (three choices: $W_{left}^{(l)}, W_{right}^{(l)}, W_{self}^{(l)}$). $h_i^{(l-1)}$ is the hidden vector from layer $l - 1$, $W^{(l)}$ and $b^{(l)}$ are the trainable matrices and biases of the W-GCN at layer $l$, LN is layer normalization, and $\sigma$ is the ReLU activation function.

Finally, we take the output $h^{(L)}$ of the $L$-th layer as the structured concept feature $\widetilde{C}$ and feed it into the language decoder, which helps to establish syntactic relationships and dependencies of texts, thereby generating more accurate text descriptions.

## 2.4 Language Decoder

With the enriched visual tokens $\widetilde{V}$ and the position-aware semantic tokens $\widetilde{C}$ from W-GCN, we integrate them into the Transformer-based decoder for sentence generation. Formally, let $Y_{gt} = \{w_0, w_1, ..., w_{T-1}\}$ denote the description (T: word number) of the input image $I$. The sentence decoder takes each word as input and learns to predict the next word auto-regressively conditioned on $\widetilde{V}$ and $\widetilde{C}$. The formulations are

$$H_i = \text{MHA}(w_i, \widetilde{V}, \widetilde{V}) + \text{MHA}(w_i, \widetilde{C}, \widetilde{C}) \quad (12)$$

$$y_{i+1} = \text{LN}(H_i + w_i) \quad (13)$$

where $y_{i+1}$ is the $(i+1)^{th}$ word of the predicted sentence, and $H_i$ is the hidden state. $Y = [y_1, y_2, ..., y_T]$ is the predicted sentence. The loss function of captioning can be defined as:

$$\mathcal{L}_{cap} = \sum_{t=1}^{T} \text{CE}(Y, Y_{gt}) \quad (14)$$

where CE is the cross-entropy loss. Thus, the total loss is the combination of visual concept prediction loss and the language prediction loss:

$$\mathcal{L} = \mathcal{L}_{cap} + \beta \cdot \mathcal{L}_c. \quad (15)$$

where $\beta$ is the hyper-parameter which aims to control the balance of two losses. To this end, our method can be trained in an end-to-end manner, which is kind to training and faster inference speed.

## 3 Experiment Settings

### 3.1 Datasets and Metrics

Our experiments are conducted on the MS COCO (Lin et al., 2014), which is the most popular image captioning benchmark dataset. It consists of more than 120,000 images, and each image is equipped with five human-annotated descriptions. We follow Karpathy's split, which divides 5,000 images for validation, 5,000 images for testing, and the rest for training (Karpathy and Fei-Fei, 2015). For fair comparison with other techniques, we leverage pycocoevalcap package to calculate five evaluation metrics: BLEU-N (Papineni et al., 2002),

METEOR (Denkowski and Lavie, 2014), ROUGE (Lin, 2004), CIDEr (Vedantam et al., 2015), and SPICE (Anderson et al., 2016).

### 3.2 Implementation Details

Our feature extractor is CLIP (Mokady et al., 2021) with the ResNet-101 (He et al., 2016) backbone and the dimension of the grid visual feature is 2048. Following previous work (Li et al., 2022b), to build concept vocabulary, we filter out low-frequency words and convert all uppercase letters to lowercase letters to all caption descriptions. Thus, a concept vocabulary that containing 906 words is constructed. The word distance in the pre-constructed lexicon $D$ is 3. The number of layers in Weighted GCN is set to 2. The Transformer block in the Feature Encoder Module, the Concept Prediction Module and the Language Prediction Module are 3 layers, 6 layers, and 6 layers, respectively. The size of the hidden state features is set to 512. The query size is set to 17. $\beta$ is set to 1 in this work. Our code is developed based on the COS-Net[1].

The model is trained using a typical two-stage training method. In the first stage, we utilize Adam optimizer and cross-entropy loss with a learning rate of 0.0005 and take about one hour per epoch. In the second stage, the self-critical sequence training strategy is used to further optimize the CIDEr scoring model, and the learning rate is set to 0.00005, which takes about 4 hours per epoch. In inference, the beam size is set to 3. The number of parameters our model used is 20M. All experiments are conducted on a single RTX 3090.

## 4 Results and Analysis

### 4.1 Main Results

Our main results are shown in Table 1 and Table 2. All compared methods can be briefly grouped into two kinds. Firstly, the conventional methods, e.g., Up-Down, M2 Transformer, X-Transformer, which utilize the pre-trained Faster R-CNN (backbone: ResNet-101) to extract visual inputs. The second kind approaches, e.g. CTX+M2, which take the strong CLIP (backbone: ResNet-101) grid features as visual inputs. Compared with these previous studies, the results of our method have improved in each evaluation metrics, especially in CIDEr. Specifically, our method is 11.4 higher than the classic GCN-LSTM method (GCN-based model),

---

[1]https://github.com/YehLi/xmodaler/tree/master/configs/image_caption/cosnet

| Method | Cross Entropy | | | | | |
| | BLEU-1 | BLEU-4 | METEOR | ROUGE | CIDEr | SPICE |
|---|---|---|---|---|---|---|
| Up-Down(Anderson et al., 2018) | 77.2 | 36.2 | 27.0 | 56.4 | 113.5 | 20.3 |
| GCN-LSTM(Yao et al., 2018) | 77.3 | 36.8 | 27.9 | 57.0 | 116.3 | 20.9 |
| AoANet(Huang et al., 2019) | 77.4 | 37.2 | 28.4 | 57.5 | 119.8 | 21.3 |
| X-Transformer(Pan et al., 2020) | 77.3 | 37.0 | 28.7 | 57.5 | 120.0 | 21.8 |
| ViTCAP (Fang et al., 2022) | - | 35.7 | 28.8 | 57.6 | 121.8 | 22.1 |
| **(Ours)** | **78.3** | **38.6** | **29.3** | **58.5** | **125.3** | **22.4** |

Table 1: Comparison with the state-of-the-art methods on COCO Karpathy test split in the first stage, namely Cross Entropy.

| Method | CIDEr Optimization | | | | | |
| | BLEU-1 | BLEU-4 | METEOR | ROUGE | CIDEr | SPICE |
|---|---|---|---|---|---|---|
| Up-Down(Anderson et al., 2018) | 79.8 | 36.3 | 27.7 | 56.9 | 120.1 | 21.4 |
| GCN-LSTM(Yao et al., 2018) | 80.5 | 38.2 | 28.5 | 58.3 | 127.6 | 22.0 |
| AoANet(Huang et al., 2019) | 80.2 | 38.9 | 29.2 | 58.8 | 129.8 | 22.4 |
| M2 Transformer(Cornia et al., 2020) | 80.8 | 39.1 | 29.2 | 58.6 | 131.2 | 22.6 |
| X-Transformer(Pan et al., 2020) | 80.9 | 39.7 | 29.5 | 59.1 | 132.8 | 23.4 |
| RSTNet (Zhang et al., 2021b) | 81.1 | 39.3 | 29.4 | 58.8 | 133.3 | 23.0 |
| DLCT (Luo et al., 2021) | 81.4 | 39.8 | 29.5 | 59.1 | 133.8 | 23.0 |
| ReFormer (Yang et al., 2022) | 82.3 | 39.8 | 29.7 | 59.8 | 131.9 | 23.0 |
| ViTCAP (Fang et al., 2022) | - | 40.1 | 29.4 | 59.4 | 133.1 | 23.0 |
| CTX+M2 (Kuo and Kira, 2022) | 81.5 | 39.7 | 30.0 | 59.5 | 135.9 | 23.7 |
| DIFNet (Wu et al., 2022) | 81.7 | 40.0 | 29.7 | 59.4 | 136.2 | 23.2 |
| **Ours** | **82.6** | **41.5** | **30.2** | **60.2** | **139.0** | **24.2** |

Table 2: Comparison with the state-of-the-art methods on COCO Karpathy test split in the second stage, namely Cider Optimization.

and 7.1 higher than the recent ReFormer method. This is due to our approach considering both visual and textual semantics and finding ways to close the semantic gap between them. In the CP module, the extraction concept distills the textual information. And we use attention map convolutional networks in the W-GCN module to explore the structural relationships between concepts, which has been completely ignored in previous studies. It is important that when initializing the graph, we fully consider the context information and firmly glue the strongly related word pairs together, so that the generated sentences are more human-like. We further test the running speed with other state-of-the-art methods. (Nguyen et al., 2022) runs at 138 ms/image, (Cornia et al., 2020) runs at 178 ms/image, and the pre-trained based method (Zhang et al., 2021a) runs at 542 ms/image. Our method achieves 214 ms/image running speed, which illustrates the effectiveness of our method. Meanwhile, the results show that the proposed W-GCN module and CP

module have little effect on the running speed.

## 4.2 Qualitative Results

To qualitatively illustrate the effectiveness of our proposed approach and make the results more convincing, we present and analysis some qualitative results here. Firstly, some examples are shown in Figure 3, where **GT** is the ground-truth sentence labeled by human and **Ours** is the predicted result of our model. It can be seen that our model accurately describes the content of the image, and the words are appropriate, logical, and elegant, reaching the human level. It captures the key semantic words, and generated sentence follow the same patterns as the GT.

In addition, we present a structured graph of the concept to illustrate the value of the W-GCN module in Figure 4. Nodes represent concepts, and the edges indicate the strength of the relationship between the two concepts. It can be seen that, first, our method can predict semantic concepts

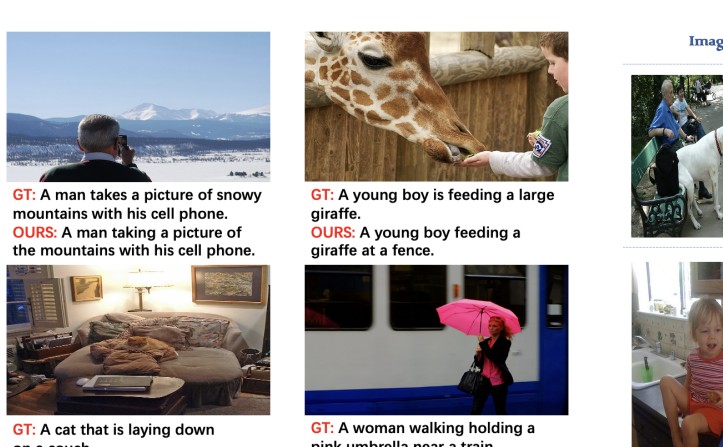

Figure 3: The qualitative results of our approach. Our approach is good at capturing the relations between semantic concepts, improving the preformance.

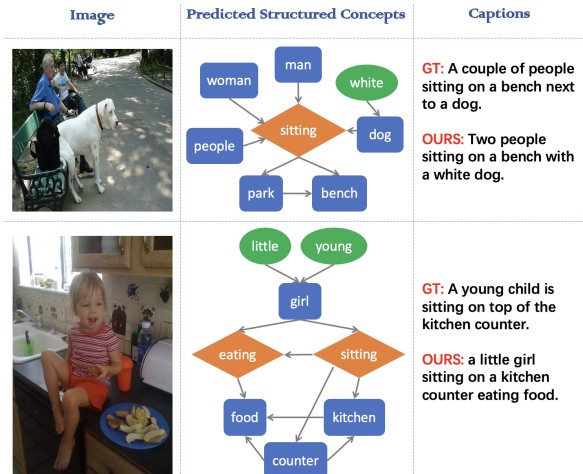

Figure 4: Visualization results of the procedure of our method, including the structured concepts prediction. It is not difficult to find that the structured concepts we predict are of good quality.

well, including key nouns, verbs and adjectives in the sentence. Meanwhile, in the absence of supervision information, our W-GCN can capture the relations between semantic concepts, including adjectives pointing to the correct nouns, and nouns pointing to the correct verbs. For example, in the second case, our model can understand the "little→girl", "girl→eatting" and "eatting→food". The predicted structured concepts further strengthens our sentence generation.

### 4.3 Ablation Studies

In order to explore the effect of each proposed module, we perform a series of ablation experiments. For the Weighted GCN Module, we also conduct several experiments to verify the effect of different graph construction on the results. All the results are recorded in Table 3.

#### 4.3.1 Effect of the Proposed Modules

"**Baseline**" is the simple Transformer encoder-decoder structure. "**Baseline+CP**" represents the "**Baseline**" combine with the concept prediction module. "**Baseline+CP+WGCN(Ours)**" is our best model, which integrate W-GCN on the basis of "**Baseline+CP**"[2].

From the results, it can be seen that, first, the two proposed modules "concept prediction" and "Weighted GCN" are both effective and greatly improve the captioning performance. When comparing BLEU-1 metric, we have an interesting observation that "**Baseline+CP**" is slightly worse than

---

[2]Since our W-GCN is built upon concept prediction, we cannot conduct "**Baseline+W-GCN**" experiment.

"**Baseline+CP+WGCN(Ours)**", which means, no matter whether we learn the relations between concepts or not, our concept predictions are accurate to capture the semantic words of the sentence. The accuracy of this keyword prediction will lead to a good result in BLEU-1 metric. Thus, there is not much difference in BLEU-1 metric between the two methods. But the results of other metrics can show the superiority of our W-GCN.

#### 4.3.2 Discussion on Graph Construction

As aforementioned, graph construction is equivalent to the initial the adjacency matrix, and we conduct a variety of experiments to verify the effect of the initial graph constructed by our word dependencies.

"**Random**" indicates that when initializing the graph, the interrelationships among concepts are ignored, and the corresponding adjacency matrix is a randomly generated 0-1 matrix. "**1-for-all**" indicates that when the graph is initialized, all concepts are considered to be related, and the elements in the adjacency matrix are all set to 1. "**MLP**" learns the relationship between concepts. Each concept is affected by other concepts to different degrees, which is learned by MLP. "**Threshold-N**" means that the initial graph is built by the PMI scores of word dependencies, where N is the threshold. For all concepts of an image, we calculate the PMI score between pairs. If the PMI score is greater than the threshold, the corresponding element of the adjacency matrix is set to 1, otherwise to 0.

| Discussion | Method | B-1 | B-4 | M | R | C | S |
|---|---|---|---|---|---|---|---|
| Proposed Modules | Baseline | 77.7 | 37.7 | 28.8 | 57.8 | 121.7 | 21.8 |
| | Baseline+CP | 78.2 | 38.0 | 29.0 | 58.0 | 123.4 | 22.1 |
| | Baseline+CP+WGCN(**Ours**) | **78.3** | **38.6** | **29.3** | **58.5** | **125.3** | **22.4** |
| Graph Construction | Random | 78.1 | 38.0 | 29.2 | 58.1 | 123.6 | 22.2 |
| | 1-for-all | **78.6** | 38.4 | 29.2 | 58.2 | 124.4 | 22.4 |
| | MLP | 77.7 | 38.1 | 29.1 | 58.0 | 123.7 | 22.1 |
| | Threshold-0.1 | 78.5 | 38.6 | 29.2 | 58.3 | 124.4 | 22.2 |
| | Threshold-0.3 | 78.5 | 38.4 | 29.1 | 58.4 | 124.1 | 22.1 |
| | Threshold-0.5(**Ours**) | 78.3 | 38.6 | **29.3** | **58.5** | **125.3** | **22.4** |
| | Threshold-0.7 | 78.6 | **39.0** | 29.2 | 58.2 | 124.6 | 22.3 |

Table 3: The results of ablation studies (Cross Entropy). Here, B-N, M, R, C, and S represent BLEU-N, METEOR, ROUGE, CIDER, SPICE, respectively.

The results are in Table 3, and from the results, we have several observations. First of all, the results of "**Random**" and "**MLP**" are the worst. It can be seen that the relations between concepts is not random and should be established based on some word dependencies. Secondly, the effect of "**1-for-all**" is not good enough. It considers all concepts to be related. Obviously, this way of handling doesn't make sense, and there is no relations between some semantic words, such as "red" and "grass". Last, for word dependencies, it can be seen that the effect reaches best when the threshold is set to 0.5.

## 5  Related Work

Image Captioning is a practical task that has been widely concerned and studied. There have been many studies before (Liu et al., 2019; Guo et al., 2020; Chen et al., 2021; Song et al., 2023b,a; Liu et al., 2021; Nie et al., 2021; Tu et al., 2021). Inspired by the Transformer structure in NLP, the Transformer-based encoder-decoder structure has recently become mainstream, where the interaction between multi-modal information can be enhanced. (Herdade et al., 2019; Li et al., 2022a) integrates the spatial relationships between objects through geometric attention based on the Transformer structure. (Nguyen et al., 2022) use a set of learnable object queries to extract semantics from multi-scale features, and feed them into a Transformer to generate captions.

The previous researches have shown that exploring semantic concepts and their relations make con-

tributions to high-quality descriptions. (Chen et al., 2020) encode image into Abstract Scene Graphs, which represent semantics and structure at a fine-grained level, thus generating detailed descriptions. (Zhang et al., 2022) not only construct a multi-modal relational graph for images, but also encodes relational graphs for all sentences in the dataset to fully capture language features. Then a cascaded GAN is developed to achieve cross-domain alignment of image and text pairs, which are fed into the decoder. (Shi et al., 2020) explore the semantics available in captions and construct a caption-guided visual relationship graph, and leverage it to enhance image representation and caption generation. The Concept Token Network is introduced to predict concept words based on visual information. These concepts contain rich semantic information, which benefits the image captioning task (Fang et al., 2022; Li et al., 2022b). (Zeng et al., 2022) try to capture the hierarchical semantic structure in the text space, helps visual features carry semantics and generates finer-grained and more reasonable phrases and collocations. To effectively extract contextual information, constructing a graph is proposed to model the relations between words (Tian et al., 2020).

## 6  Conclusion

In this paper, we explore the significance of concepts and their structures for accurately describing images in the image captioning task. To predict concepts and their structures, we propose a structured concept predictor (SCP), so that enhance the

contribution of visual signals and further use their relations to distinguish cross-modal semantics for better description generation. Particularly, we design weighted graph convolutional networks (W-GCN) to depict concept relations driven by word dependencies, and then learns differentiated contributions from these concepts for following decoding process. Therefore, our approach captures potential relations among concepts and discriminatively learns different concepts, so that effectively facilitates image captioning with inherited information across modalities. Extensive experiments indicate that our approach achieves competing performances on MS COCO benchmark. Qualitative analyses confirm its ability in better capturing structural relationship among semantic concepts, so that offers good interpretability for the proposed model.

## Limitations

When predicting structured concepts in this paper, there is a premise, that is, predict concepts at first. However, when constructing the semantic concept vocabulary in this paper, we follow the existing work (Li et al., 2022b). The semantic concept vocabulary is obtained only by analyzing the corpus and the frequency of words. In future work, we can consider semantic concepts based on n-grams in the corpus. Secondly, we find that in our experiment, for some samples, our model can only predict couple concepts, which make the structured concept prediction meaningless. In future work, we aim to establish a better cross-modal correspondence, so that more concepts can be predicted.

## Ethics Statement

This paper proposed a new approach for Image Captioning and use existing datasets for evaluation. We do not foresee any ethic issues of this study.

## Acknowledgements

This work is supported by the National Natural Science Foundation of China under Grant 62302474 and the National Science Fund for Excellent Young Scholars under Grant 62222212.

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
