# OpenReview forum: "Improving Image Captioning via Predicting Structured Concepts"
_EMNLP/2023/Conference — EMNLP 2023 Main_

### Official Review · Reviewer_y9d1 · 2023-08-02

**Typos Grammar Style And Presentation Improvements:** n.a.
**Soundness:** 3

**Excitement:**

3: Ambivalent: It has merits (e.g., it reports state-of-the-art results, the idea is nice), but there are key weaknesses (e.g., it describes incremental work), and it can significantly benefit from another round of revision. However, I won't object to accepting it if my co-reviewers champion it.

**Missing References:**

n.a.

**Paper Topic And Main Contributions:**

A structured concept predictor (SCP) is proposed to predict concepts and their structures. The researchers integrate these predictions into captioning, aiming to enhance the contribution of visual signals in the task using concepts. Additionally, they leverage the relations among these concepts to distinguish cross-modal semantics for improved description generation. Specifically, the authors employ weighted graph convolutional networks (W-GCN) to depict concept relations based on word dependencies. This enables the model to learn differentiated contributions from the identified concepts during the decoding process. As a result, their approach successfully captures potential relations among concepts and discriminatively learns different concepts, effectively facilitating image captioning by leveraging information across modalities. The effectiveness of the proposed approach, along with each module introduced in this research, is demonstrated through extensive experiments and their corresponding results.

**Questions For The Authors:**

Why do the authors test the method only on captioning tasks? I think the method can also be effective in VQA tasks.

**Reasons To Accept:**

- In this paper, a structured concept predictor (SCP) is proposed, which is utilized to enhance image captioning. The idea of incorporating structured information between visual concepts is deemed valid and crucial.

- The proposed method demonstrates superior performance compared to strong baselines in captioning tasks.

- To verify the effectiveness of the proposed method, the authors conduct ablation studies.

**Reasons To Reject:**

- The method proposed in this study involves several complex modules, such as GCN and the concept predictor. Including an analysis of the inference speed in comparison to other baselines would add strength to the research.

- The writing quality could be improved. For instance, there were difficulties in understanding parts of L172, and it might be helpful if the authors added a "However" before that section.

- Regarding the query embedding, more explanation is needed. Are they learnable queries, similar to BLIP2? The authors should provide additional details about the modules to clarify their implementation and functioning.

**Reproducibility:**

4: Could mostly reproduce the results, but there may be some variation because of sample variance or minor variations in their interpretation of the protocol or method.

**Reviewer Confidence:**

3: Pretty sure, but there's a chance I missed something. Although I have a good feel for this area in general, I did not carefully check the paper's details, e.g., the math, experimental design, or novelty.

---

> ### Author Rebuttal · Authors · 2023-08-29
>
> Thanks for your sincere review. We will carefully revise our paper following your suggestions.
>
> To the Rejections:
>
> To R1:
>
> Thanks for your kind suggestions. We tested the running speed of our method, which is 214 ms/image. It is acceptable. We also ran the speed of some comparison methods, which are listed as follows:
>
> GRIT (ECCV 2022): 138 ms/image
>
> M2 Transformer: 178 ms/image
>
> Vin VL_large: 542 ms/image
>
> The GCN and concept predictor have little influence on the running speed of our method. Our method is still significantly faster than the pre-trained based method. We will add the discussion in the revised paper.
>
> To R2:
>
> Thanks for pointing out our writing typos. We will carefully check the paper, and make sure there are no writing typos and logical problems in the revised paper.
>
> To R3:
>
> Sorry for the confusion. We will add more details in the revised paper to introduce query embeddings. The learnable query in our paper is similar to BLIP2. However, our query learns the essential concepts within the images. Through the image interaction, each of our learnable queries focuses on a specific area of the image and learns the information (concepts) contained in the image. These concepts include objects, relative positions between objects, actions, etc.
>
> To the question:
>
> Thanks for your suggestions. Our method can indeed be expanded on tasks such as VQA, but this paper focuses on the image captioning task and its challenges, and the extensive method in VQA would be slightly different. In the revised paper, we will add more discussion of the scalability of our method, and in future work, we will add more experiments of different tasks to explore the scalability of the method.
>
> On the VQA task, we can capture the structured concepts of the question (Q) and the structured concepts of the image (V), and then, we can leverage the multi-modal graph convolutional networks to merge the two graphs to generate answers (A) better. Slightly different from our current method, which only leverages structured concepts of the image in captioning.

---

### Official Review · Reviewer_jqw3 · 2023-08-03

**Typos Grammar Style And Presentation Improvements:** None,
**Soundness:** 3

**Excitement:**

3: Ambivalent: It has merits (e.g., it reports state-of-the-art results, the idea is nice), but there are key weaknesses (e.g., it describes incremental work), and it can significantly benefit from another round of revision. However, I won't object to accepting it if my co-reviewers champion it.

**Missing References:**

[1] Semantic Compositional Networks for Visual Captioning, CVPR 2017.

[2] Adaptively Attending to Visual Attributes and Linguistic Knowledge for Captioning, ACM MM 2017.

[3] Fused GRU with semantic-temporal attention for video captioning, Neurocomputing 2020.

[4] R^3Net: Relation-embedded Representation Reconstruction Network for Change Captioning, EMNLP 2021.

**Paper Topic And Main Contributions:**

This paper proposes a structured concept predictor to predict concepts and their structures, and integrating these concepts into the decoder for caption generation. Besides, considering that the contribution of different nodes may be different, this paper further designs a weighted GCN to not only construct the connections between different concept nodes, but also assign different weights to these connections. The weighted average concept features are called structured concept features.  Finally, the structured concept features and visual features are fed into a transformer decoder for caption generation. Extensive experiments show that the proposed method achieves SOTA results on the MS COCO dataset.

**Questions For The Authors:**

1. In L83-85, the authors said that“drink” is more likely to be followed by “water” based on the linguistic prior, which I agree with. However, in the first case of Fig.1, the authors argued that “eating” is followed by “apple” also based on the linguistic prior. This is not convincing to me. Please provide more explanation or change an example.

**Reasons To Accept:**

1. I appreciate the idea of introducing concepts and modeling their relations, which is key to generate high-quality image captions.
2. The proposed method achieves state-of-the-art results on a well-known dataset.

**Reasons To Reject:**

1.	There is a lack of comparison with the previous work that addresses the same problem.  Specifically, in Sec. Introduction L53-64, the authors claimed that extracted visual information is often insufficient, so previous methods rely too much on linguistic priors for caption generation. This claim is similar
to the previous CVPR work, DIFNet: Boosting Visual Information Flow for Image Captioning (CVPR’22). However, the authors did not make comparison with it in terms of method and experiment. In my opinion, this is not rigorous and not beneficial to highlight the contribution of this work.
2.	Missing references. The major contribution of this paper is to predict concepts from ground-truth captions through a multi-label classification task. This idea has been tried by some of the previous works [1-4]. However, the authors did not cite and discuss them.

[1] Semantic Compositional Networks for Visual Captioning, CVPR 2017.

[2] Adaptively Attending to Visual Attributes and Linguistic Knowledge for Captioning, ACM MM 2017.

[3] Fused GRU with semantic-temporal attention for video captioning, Neurocomputing 2020.

[4] R^3Net: Relation-embedded Representation Reconstruction Network for Change Captioning, EMNLP 2021.

3. Insufficient experiment. This paper lacks the experiment on the MS-COCO online testing server, which has been widely done in most previous works. Besides, in L356-357, the word distance in the pre-constructed lexicon D is 3. The authors might consider providing analysis for this hyper-parameter.

**Reproducibility:**

3: Could reproduce the results with some difficulty. The settings of parameters are underspecified or subjectively determined; the training/evaluation data are not widely available.

**Reviewer Confidence:**

4: Quite sure. I tried to check the important points carefully. It's unlikely, though conceivable, that I missed something that should affect my ratings.

---

> ### Author Rebuttal · Authors · 2023-08-29
>
> Thanks for your sincere review. We will carefully revise our paper following your suggestions.
>
> To the Rejections:
>
> To R1:
>
> Thanks for your kind suggestions. We will cite this paper and add a comparison in the revised paper.
>
> Similar to the DIFNet (CVPR 2022) that you pointed out, we also focus on the same challenge. Differently, we solve this challenge from the perspective of natural language processing. First, we will predict the concept words, and then, we leverage the prior knowledge of concept word dependencies to capture the structured concepts in order to guide the generation of trivial words. The DIFNet adds additional visual information (segmentation image) to obtain strong visual features and increase the contribution of vision to generate captions. DIFNet and our method solve this challenge from two perspectives, and our method outperforms DIFNet in metrics.
>
> To R2:
>
> Thanks for pointing out our weakness. We will cite these papers and discuss them. Besides, our method goes a step further than these methods, as our method learns to capture the relations between concepts from the perspective of NLP. Our method proposes weighted graph convolutional networks and leverages the prior knowledge of concept word dependencies to learn the relations between concepts, which would further encourage language generation to be more reasonable.
>
> To R3:
>
>  Thanks for your suggestions.
>
> I. We have done online test of our method. The result is slightly lower than the offline test, which is also in line with the results on COS-Net (CVPR 2022). The online test results are lower than the offline test in COS-Net. It is worth noting that our code is developed based on COS-Net. Meanwhile, our online results are still better than the baselines that we compared in the submitted paper, such as COS-Net (CVPR 2022), ReFormer (ACM-MM 2022) and multiple baselines.The results of our method are listed as follows:
>
> cider-c5: 1.368; cider-c40: 1.381
>
> We will add the results to the revised paper, and thanks for your suggestions.
>
> II. Thanks for your suggestion. We will supplement the discussion of D in the revised paper. When D<=2, the concepts basically have no initial relations, resulting in the initial matrix being almost equal to the all-0 matrix, making it difficult to learn relations between concepts. The results drop a lot when D<=2. The results in D=4 are similar to D=3. The performance when D=3 is slightly better. When D>=5, almost all concepts have initial relations, and the performance drops a lot. Thus, we choose D=3 in the submitted paper.
>
> To the Question:
>
> We will change the example in the revised paper, and thanks for your sincere suggestion.

---

### Official Review · Reviewer_hF5e · 2023-08-05

**Soundness:** 3

**Excitement:**

3: Ambivalent: It has merits (e.g., it reports state-of-the-art results, the idea is nice), but there are key weaknesses (e.g., it describes incremental work), and it can significantly benefit from another round of revision. However, I won't object to accepting it if my co-reviewers champion it.

**Paper Topic And Main Contributions:**

The manuscript delves into the challenging task of bridging the semantic gap between images and texts for image captioning. The authors introduce the structured concept predictor (SCP) to predict both concepts and their relationships, then incorporate these into the image captioning. They utilize weighted graph convolutional networks (W-GCN) to depict relations among concepts influenced by word dependencies. By doing so, they aim to improve the understanding of cross-modal semantics and, consequently, the quality of generated image captions. Their results, supported by extensive experimentation, showcase the effectiveness of the proposed approach.

**Questions For The Authors:**

1. In the process of predicting concepts and their structures, what challenges did you encounter? And how did you mitigate them?
2. How does your method compare to other existing techniques, not just in terms of performance but also in terms of computational efficiency and complexity?
3. Can the SCP be modified or adapted for tasks other than image captioning? If yes, what potential challenges do you foresee?
4. In implementing the W-GCN, how did you decide upon the weighting mechanism? Was there an iterative process or any fine-tuning involved?
5. It would be insightful to understand more about how you see the future developments or potential extensions of this method. Are there any immediate enhancements you're considering?

**Reasons To Accept:**

1. The proposed approach is sound and well-motived.
- The manuscript introduces an innovative approach by predicting not just the concepts but also their inter-relationships. This structured concept prediction can offer significant improvements in understanding cross-modal semantics.

2. The proposed W-GCN to capture concept relations and dependencies is interesting.

3. The results prove the effectiveness of the proposed method.

**Reasons To Reject:**

1. The paper doesn't sufficiently engage with related works, potentially missing out on drawing distinctions or acknowledging overlaps with other scene graph based methods, e.g., [1][2].

2. A comprehensive evaluation, including a performance metric such as inference speed, might provide a more rounded understanding of the method's practicality.

3. A broader range of applications or tasks, apart from just captioning, might provide a better understanding of the versatility and adaptability of the proposed method.

4. Several experiments are missing.
- Error analysis.
- MSCOCO online evaluation.
- Although the proposed did not achieve state-of-the-art performances, it is still necessary to report the state-of-the-art results in the Tblaes. It can better help the readers understand the strengths and weaknesses of the proposed approach.

[1] R^3Net: Relation-embedded Representation Reconstruction Network for Change Captioning, EMNLP 2021.
[2] DIFNet: Boosting Visual Information Flow for Image Captioning, CVPR 2022.

**Reproducibility:**

4: Could mostly reproduce the results, but there may be some variation because of sample variance or minor variations in their interpretation of the protocol or method.

**Reviewer Confidence:**

5: Positive that my evaluation is correct. I read the paper very carefully and I am very familiar with related work.

---

> ### Author Rebuttal · Authors · 2023-08-29
>
> Thanks for your carefully reviewing and detailed suggestions. It would improve the quality of our paper a lot.
>
> To the Rejections:
>
> To R1:
>
> Thank you for your suggestions. We will cite and discuss these two papers.
>
> Besides, our method is different from the scene graph-based approach. The scene graph is established based on the predicting visual objects and their relationship. The prediction may make errors since images vary and their object relations vary. Meanwhile, the scene graph also has a gap between language descriptions.
>
> From the perspective of natural language processing, our concept relationship is learned based on concept word dependencies, which can more accurately capture the concept relationship, and structured concepts have no gap between language descriptions. Thus, our method has a good performance.
>
> To R2:
>
> Thanks for your kind suggestions. We tested the running speed of our method, which is 214 ms/image. It is acceptable. We also ran the speed of some comparison methods, which are listed as follows:
>
> GRIT (ECCV 2022): 138 ms/image
>
> M2 Transformer: 178 ms/image
>
> Vin VL_large: 542 ms/image
>
> The GCN and concept predictor have little influence on the running speed of our method. Our method is still significantly faster than the pre-trained based method. We will add the discussion in the revised paper.
>
> To R3:
>
> Thanks for your suggestions. Our method can indeed be expanded on tasks such as VQA, but this paper focuses on image captioning tasks and their challenges, and the modified method in VQA would be slightly different. In the revised paper, we will add more discussion of the scalability of our method, and in future work, we will add more experiments of different tasks to explore the scalability of the method.
>
> On the VQA task, we can capture the structured concepts of the question (Q) and the structured concepts of the image (V), and then, we can leverage the multi-modal graph convolutional networks to merge the two graphs to generate answers (A) better. Slightly different from our current method, which only uses structured concepts of the image in captioning.
>
> To R4:
>
> Thanks for your suggestions.
>
> I. We discuss the limitations at the end of the submitted paper. We will substitute it into the error analysis in the revised paper. We find that in our experiment, for some samples, our model can only predict a couple of concepts, which makes the structured concept prediction meaningless.
>
> II. We have done the online test of our method. The result is slightly lower than the offline test, which is also in line with the results on COS-Net (CVPR 2022). The online test results are lower than the offline test in COS-Net. It is worth noting that our code is developed based on COS-Net.
>
> Meanwhile, our online results are still better than the baselines that we compared in the submitted paper, such as COS-Net (CVPR 2022), ReFormer (ACM-MM 2022), and multiple baselines.
>
> The results of our method are listed as follows:
> cider-c5: 1.368; cider-c40: 1.381
>
> We will add the results to the revised paper, and thanks for your suggestions.
>
> III. Thanks for your kind suggestions. We will supplement the results of SOTA methods in the paper and analyze our shortcomings.
>
> It is worth noting that our main results are conducted on a single RTX3090, and most SOTA methods are conducted on 8*A100.
>
> To the Questions:
>
> To Q1:
>
> During the experiment, the biggest challenge is how to capture the relationship between concepts. We propose and try a variety of methods, and finally, we propose weighted graph convolutional networks and leverage the prior knowledge of concept word dependencies, so as to accurately capture the concepts and their relations, resulting in better performance.
>
> To Q2:
>
> All our experiments are carried out on a single RTX3090, and the performance of our method reaches SOTA. It is worth noting that most of the SOTA methods are conducted on 8*A100. We also test the running speed, which is 214 ms/image. It is acceptable. As we mentioned, we have a comparable running speed, and our method is still significantly faster than the pre-trained based method.
>
> To Q3:
>
> SCP can be modified to apply in the VQA task. On the VQA task, we can capture the structured concepts of the question (Q) and the structured concepts of the image (V), and then, we can leverage the multi-modal graph Convolutional networks to merge the two graphs to generate answers (A) better. Slightly different from our current method, which only leverages structured concepts of the image in captioning. The potential challenge is how to align the structured concepts of the question (Q) and the structured concepts of the image (V).
>
> To Q4:
>
> The initial weight is established based on word dependencies, which is also called point-wise mutual information (PMI) of any concept. We have tried a variety of initial weights, and word dependencies lead to the best results. The most likely reason is that word dependencies are the prior knowledge of the dataset, which helps the model understand natural language better. The subsequent relations are learned through the graph convolutional networks, which are powerful for learning the relations between nodes.
>
> To Q5:
>
> We are considering predicting the exact position of the concept in the sentence, and the subsequent text generation can be seen as a process of filling in the blanks. This process of filling in the blanks can refer to bi-directional (forward and backward) concepts and their relations. In this way, during the text generation process, it does not rely too much on word priors, like "drink" followed by "water" in any case, resulting in image-related sentences.

---

### Official Review · Reviewer_uJ4d · 2023-08-06

**Soundness:** 4

**Excitement:**

4: Strong: This paper deepens the understanding of some phenomenon or lowers the barriers to an existing research direction.

**Missing References:**

No missing references.

**Paper Topic And Main Contributions:**

This paper proposes incorporation of concept structures that capture relationships between concepts through a weighted graph convolutional network, into image captioning. The authors build a convincing framework using among others a mutual information based approach, and demonstrate significant advancement of the state of the art in image captioning across multiple evaluation criteria or measures.

**Questions For The Authors:**

1. Do you have any intuitions as to what kind of content would benefit most from your proposed new technique?

**Reasons To Accept:**

1. Solid literature review.
2. Insightul and sound new approach to incorporation of concept structures worked out thoroughly.
3. Clear improvement over the state of the art.
4. Well written paper.

**Reasons To Reject:**

1. No significant weaknesses.

**Reproducibility:**

5: Could easily reproduce the results.

**Reviewer Confidence:**

5: Positive that my evaluation is correct. I read the paper very carefully and I am very familiar with related work.

**Typos Grammar Style And Presentation Improvements:**

Typo line 017 replace “so that enhance” with “so as to enhance”
Line 225 please replace “an stragtforward” with “a straightforward”
Line 234 please replace “Potiential” with “Potential”

these are typos I noticed. I have a feeling I might have missed some. Please go through carefully and revise.

---

> ### Author Rebuttal · Authors · 2023-08-29
>
> Thanks for your high evaluation of our paper.
>
> To the question:
>
> If describing an image requires multiple-step reasoning between concepts, our method works better since our method can grasp the relations between concepts. For example, if the predicted structured concepts are “rain->man->hold”, the most likely words behind for our method would be “umbrella”. Otherwise, the likely following word of the "hold" would be "on" for other methods. Meanwhile, if the method needs to fill the blank between the words “man” and “hold”, for our method, it is very likely that no trivial words will be filled. This is because our method leverages prior knowledge of point-wise mutual information (PMI) of any concepts. If the prior knowledge tells the model that there are no words between “man” and “hold”, the predicted sentence would not have words between “man” and “hold”.  In that case, our method also works well.
>
> To the writing typos:
>
> Thanks for pointing out our writing typos. We will carefully check the paper, and make sure there are no writing typos in the revised paper.

---

### Meta-Review · Area_Chair_w8AX · 2023-09-19

**Recommendation:** 4

**Metareview:**

**Summary**: The reviewers reached a consensus that the proposed method of using concept predictor was well-motivated and well-supported in the paper. Also, reviewers acknowledged that the paper was well-written and state-of-the-art performance is achieved. Therefore, all the reviewers are positive about the results in the paper. In the meantime, several reviewers pointed out a few missing prior works and references which the authors acknowledged during the rebuttal and mentioned they would add them in the final paper. Also, a few reviewers questioned about the inference speed comparison with other methods which was presented by the authors during the rebuttal. Overall, the reviewers seemed satisfied with the authors' responses.

**Reviewers' recommendations**: Four reviewers share that the paper is sound or strong in terms of soundness. (3 'Good's and 1 'Strong'). In terms of the excitement, the reviewers also share the same score distribution which make the paper is sound and somewhat exciting.

---

### Decision · Program_Chairs · 2023-10-07

**Decision:**

Accept-Main

**Comment:**

**Summary**: The reviewers reached a consensus that the proposed method of using concept predictor was well-motivated and well-supported in the paper. Also, reviewers acknowledged that the paper was well-written and state-of-the-art performance is achieved. Therefore, all the reviewers are positive about the results in the paper. In the meantime, several reviewers pointed out a few missing prior works and references which the authors acknowledged during the rebuttal and mentioned they would add them in the final paper. Also, a few reviewers questioned about the inference speed comparison with other methods which was presented by the authors during the rebuttal. Overall, the reviewers seemed satisfied with the authors' responses.

**Reviewers' recommendations**: Four reviewers share that the paper is sound or strong in terms of soundness. (3 'Good's and 1 'Strong'). In terms of the excitement, the reviewers also share the same score distribution which make the paper is sound and somewhat exciting.